# Risk of obstetric anal sphincter tear among primiparous women with a history of female genital mutilation, giving birth in Sweden

**Bita Eshraghi**[1]*, **Jonas Hermansson**[2], **Vanja Berggren**[3], **Lena Marions**[1]

**1** Department of Clinical Science and Education, Södersjukhuset, Karolinska Institutet, Stockholm, Sweden, **2** Department of Research, Angered Hospital, SV-Hospital Group, Gothenburg, Sweden, **3** Department of Neurobiology, Caring Science and Society and Health (NVS), Karolinska Institutet, Stockholm, Sweden

* bita.eshraghi@ki.se

**Data Availability Statement:** All relevant data are within the paper.

**Funding:** The author(s) received no specific funding for this work.

## Abstract

### Background

Female genital mutilation (FGM) includes a range of procedures involving partial or total removal of the external female genitalia. It is a harmful procedure that violates human rights of girls and women. FGM has been associated with obstetric anal sphincter injury (OASI), among other adverse obstetric complications. However, the obstetric outcomes in high-income countries are not clear. The aim of this study was to compare the risk of OASI among primiparous women, with and without a history of FGM, giving birth in Sweden.

### Method

A population-based cohort-study based on data from the Swedish Medical Birth Register during the period 2014–2018. The study included primiparous women with singleton term pregnancies. We compared the risk, using multivariable logistic regression, of our main outcome OASI between women with a diagnosis of FGM and women without a diagnosis of FGM. Secondary outcomes included episiotomy and instrumental vaginal delivery.

### Result

A total of 239,486 primiparous women with a term singleton pregnancy were identified. We included 1,444 women with a diagnosis of FGM and 186,294 women without a diagnosis of FGM in our analysis. The overall rate of OASI was 3% in our study population. By using multivariable logistic regression analysis, we found that women with a diagnosis of FGM had a significantly increased odds ratio (OR) of OASI (OR 2.69, 95%CI: 2.14–3.37) compared to women without a diagnosis of FGM. We also found an association between FGM and instrumental delivery as well as the use of episiotomy.

### Conclusion

Women with a history of FGM have an almost tripled risk of OASI in comparison with women without FGM, when giving birth in a Swedish setting. Increased knowledge and

**Competing interests:** The authors have declared that no competing interests exist.

awareness regarding FGM, and its potential health implications is crucial in order to mini-mise the risk of OASI among women with FGM giving birth in high-income countries. A limi-tation in our study is the lack of information about the specific types of FGM.

## Introduction

Female genital mutilation (FGM) is defined by the World Health Organization as "all proce-dures involving partial or total removal of the female genitalia, or other injuries to the female genital organs for non-medical reasons" [1]. FGM is a more than 2000-year-old culturally sanctioned ritual that is practiced across many cultural groups, predominantly in the sub-Saharan Africa as well as in some countries in the Middle East and Asia. The prevalence of FGM is declining globally, yet progress towards eradication is slow. However, the total number of girls subjected to FGM is still increasing due to population growth. According to the United Children's Fund, more than 200 million girls and women are living with the consequences of the practice [1–3]. The procedure performed varies by tradition and country. It ranges from removal of the clitoral glans and/or prepuce (type I), excision of the labias and clitoral glans (type II) to the most severe type (type III/infibulation) that includes cutting of the labias and suturing them together to partially cover the vaginal introitus. With or without removal of the clitoral glans and prepuce. Type IV includes all other harmful procedures to the female genital organs for non-medical reasons [1].

In 1982 Sweden banned all forms of FGM [4]. It is estimated that approximately 38,000 women and girls who have undergone FGM, reside in Sweden. Among them, 7000 are under the age of 18. The group of immigrants with the highest prevalence of FGM residing in Sweden originates from Somalia, followed by Eritrea, Ethiopia, Egypt, Gambia and Sudan [5]. The increasing number of immigrants from countries where FGM is prevalent, brings new chal-lenges to health care services in receiving countries. FGM carries several negative health conse-quences affecting women's psychological, sexual, and reproductive health [6–8]. It has been suggested that the harmful consequences are more common in type III FGM (infibulation) compared with the other types [6,9].

In pregnancy, FGM has been associated with obstetric complications, however current evi-dence is limited [10]. A systemic review in 2013 reported an association between FGM and perineal tear, prolonged labour, haemorrhage and instrumental delivery. However, no associa-tion was found for caesarean section and episiotomy [11]. A meta-analysis from 2020 observed that women who had undergone FGM were twice as likely to experience perineal tears, pro-longed labour, and episiotomy [12]. Other studies performed in African countries have also shown increased risk for caesarean section, postpartum haemorrhage, episiotomy, and peri-neal tears for these women [9,13–15].

Studies conducted in western settings show different results. In a FGM specialized hospital in Australia it was found that women with and without a history of FGM had similar obstetric outcomes, except higher risk of caesarean section among women with FGM [16]. A case-con-trol study from a specialized hospital in Switzerland showed no statistical differences between FGM patients and controls in relation to fetal outcome, maternal blood loss or duration of labour. However, in patients with FGM, emergency caesarean section and third-degree peri-neal tear occurred significantly more often than in controls [17]. A case-control study from the UK found only increased use of episiotomy among women with FGM. They found no dif-ference in mode of delivery, major blood loss, perineal trauma or fetal outcome [18].

Perineal tears are common after vaginal delivery. Among them, third- and fourth-degree anal sphincter tears (OASI) are the most serious types. Beside anal incontinence, which is one of the most frequently reported complications [19], OASI may cause urinary incontinence [20–22], pain [23] and sexual dysfunction [24]. Subsequent delivery after OASI is also associated with increased frequency of a new OASI as well as caesarean section [25,26]. Regarding possible complications following OASI, it is important to minimise the risk during delivery. Numerous studies have evaluated the risk factors for OASI, with primiparity, instrumental vaginal delivery and high infant birth weight as three of the most important predictors [27–29]. Being of African or Asian origin has shown to be another important risk factor for OASI when giving birth in Sweden [28,29]. The frequency of OASI among all primiparous women in Sweden was 4.5% in 2018 and 10.7% among women that had an instrumental vaginal delivery [30]. A Swedish register study found that women from African countries, especially from Somalia (where the frequency of infibulation is high) had an almost three-fold risk for OASI compared to women born in Sweden [31]. African women, without taking the FGM status into account, living in Sweden also had a four-fold risk of OASI when delivered with vacuum extraction [29]. To be able to improve obstetric management for all women giving birth, knowledge of potential risk factors is crucial.

The aim of the present study was to compare the risk of OASI among primiparous women, with and without a history of FGM, giving birth in Sweden.

## Materials and methods

### Study population and data collection

This nationwide cohort-study was based on data from the Swedish Medical Birth Register (MBR). The register, started in 1973, contains detailed information on 99% of all births in Sweden and its validity has been assessed as high [32,33]. The information in the MBR is collected prospectively from prenatal, delivery and neonatal centers through the electronic medical record system. The study also used data from the National Patient Register (NPR). Linkage of registers is possible by using the Swedish Personal Identity Number (PIN), which is a unique identifier for every resident in Sweden [34]. Both registers are maintained by the Swedish National Board of Health and Welfare.

Maternal and obstetric outcomes for all 239,486 primiparous women with a singleton birth in Sweden between 2014–2018 were included. We excluded all women with preterm deliveries, multiple pregnancies and caesarean section. All data was scrutinised for potential outliers, mostly related to probable manual errors.

The exposure was FGM. Diagnosis of FGM is made by medical doctors in the primary care or during pregnancy or delivery. Cases of FGM were defined as women that had received one or several of three possible diagnoses for Female Genital Mutilation (Z917, Z907, O347A) according to the 10[th] International Classification of Disease (ICD-10), either through the MBR or the NPR. There was no data on the specific type of FGM.

The code Z907 has been used after extensive vulva surgery since 1997 ("acquired absence of genitals") but it also has been used for diagnosis of FGM. In 2015 the specific code for FGM, Z917 ("female genital mutilation in one's own medical history"), was introduced. To avoid misclassification of cases with Z907, that were due to vulvectomy because of vulva precancerous/cancer conditions, we excluded women born in Sweden that had received Z907 in addition to diagnosis related to vulvar cancer (C510, C511, C512, C518, C519, D071, Z854A).

The study was approved by the Swedish Ethical Review Authority (D nr 2018/1297-21/3 (2019–02323)).

## Variables

In the MBR, the variables are reported by clinicians using a checkbox and/or the ICD-10 code. When extracting variable data, we used information from checkboxes. The main outcome variable was OASI, which consists of third- and fourth degree tear: O702, O702C, O702D, O702X (grade three, which involves the anal sphincter complex) and O703 (grade four, which extends to the mucosa). Diagnosis of perineal tears is made by midwives who usually also repair first- and second-degree tears. If OASI is suspected, it is confirmed and repaired by an obstetrician. The code for the different types of tears is generated in the patient's medical journal.

Information about maternal age (year), weight (kg in early pregnancy), height (cm) and country of origin was collected from MBR together with obstetric data on length of gestation (weeks), mode of delivery (spontaneous vaginal delivery, instrumental vaginal delivery: vacuum extraction or forceps), caesarean section and episiotomy (mediolateral or midline). Information about birth weight was collected from the same source. FGM was the main exposure variable. Third- or fourth-degree tears were merged into one outcome variable.

## Statistical analysis

To examine differences in characteristics between women with and without FGM Fisher's exact test was used. A p-value less than 0.05 was considered statistically significant. Logistic regression was used to estimate the odds ratio (OR) with 95% confidence interval (95% CI), as a measure of FGM's association with the outcome. Three different models were used in the logistic regression. The first model was adjusted for age and BMI, in the second model birth weight and instrumental delivery was added, and in the third model episiotomy was added. Logistic regression analyses were conducted for each characteristic, and interaction term for episiotomy and FGM was tested. In a sensitivity analysis, women born in countries with high prevalence of FGM [2] were examined separately to disentangle the potential confounding effect of country of origin on risk of our outcome. Statistical analyses were performed using IBM SPSS Statistics software version 27 for PC.

## Results

Including participants with vaginal singleton delivery at term (≥37 weeks of gestation), we identified 1,444 (0.8%) women with a registered diagnosis of FGM and 186,294 women without a registered diagnosis of FGM as described in Fig 1.

Maternal and infant characteristics together with obstetric interventions and outcomes are shown in Table 1.

In the cohort, the overall rate of vaginal non-instrumental delivery was 86.8%, episiotomy 10.4% and OASI 3%. A significantly higher proportion of women with FGM underwent instrumental delivery and episiotomy compared to controls. The most frequent countries of origin are shown in Table 2.

Proportion of and Ors for OASI by maternal characteristics and obstetric interventions are presented in Table 3. Birthweight >4000 g and instrumental delivery were strongly related to OASI.

Table 4 presents the crude and adjusted OR for the association between FGM and OASI. Included covariates were maternal age, BMI, birthweight, instrumental delivery and episiotomy. The results from the multivariable logistic regression analysis showed that women with a diagnosis of FGM had a significantly increased OR of OASI (OR 2.69, 95% CI: 2.14–3.37) compared to women without a diagnosis of FGM. An interaction term for episiotomy and FGM was tested in the last model in Table 4, but was not statistically significant (P = 0.75) and did not change the estimates for the association between FGM and OASI (data not shown).

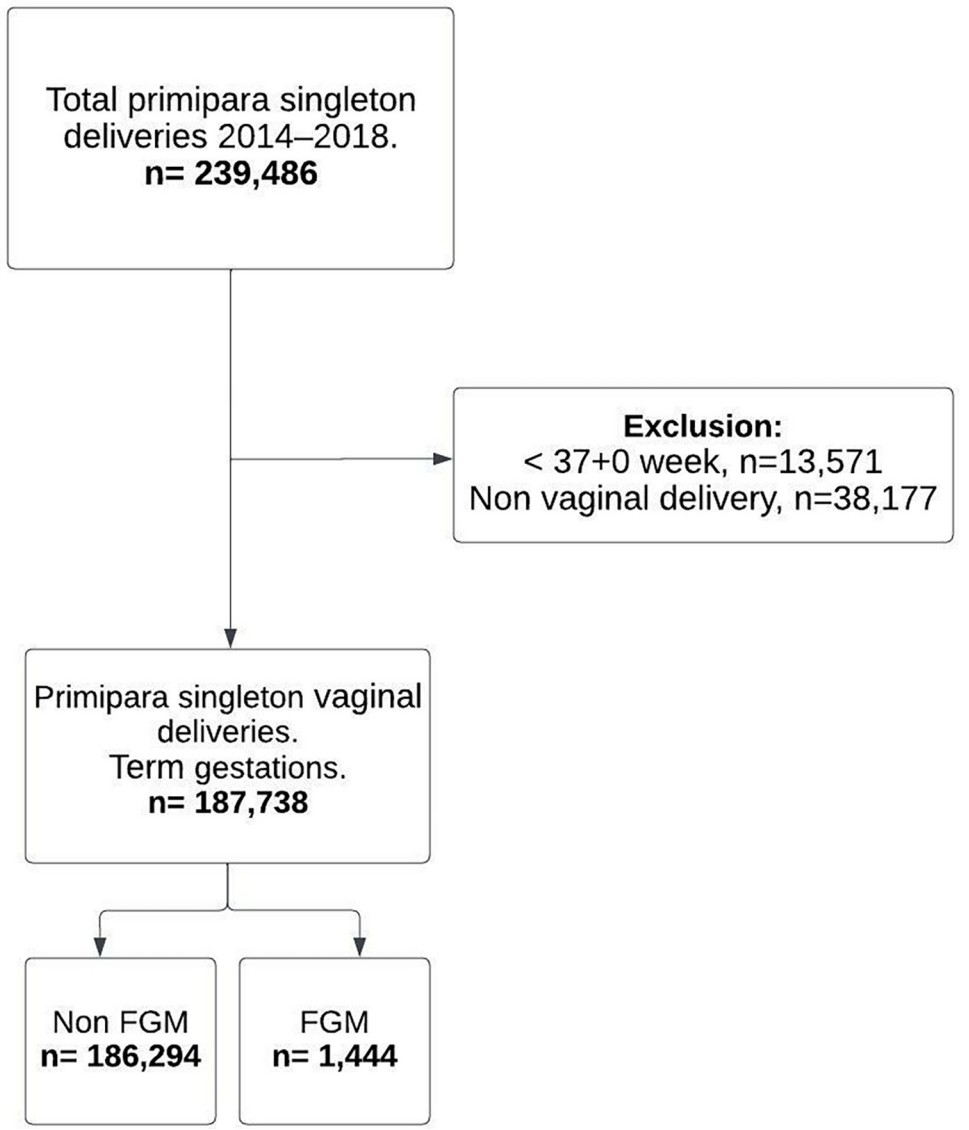

**Fig 1. Enrolment of participants from the Swedish Medical Birth Register (MBR).**

In our sensitivity analysis, the association between FGM and OASI was examined in a sub cohort of women born in countries with high prevalence of FGM [2]. Table 5 presents the crude OR and adjusted OR for OASI for this association. The OR for OASI among women with diagnosis of FGM compared to women without diagnosis of FGM in this sub cohort were in general attenuated compared to the ORs in the larger cohort.

## Discussion

Our study shows that primiparous women with a history of FGM have an almost three-fold risk for OASI when giving birth in Sweden, compared with primiparous women without a history of FGM. This confirms previous results from a Swedish register study demonstrating that women born in Somalia, where the frequency of infibulation is high, had an almost three-fold risk for OASI when giving birth compared to women born in Sweden [31]. A limitation of that

**Table 1. Maternal and obstetric characteristics of primiparous women with term vaginal deliveries by female genital mutilation (FGM) status.**

| | | FGM (n = 1444) | | Non FGM (n = 186 294) | | P-value* |
|---|---|---|---|---|---|---|
| | | n | % | n | % | |
| Maternal age, years | <25 | 656 | 45.4 | 40 034 | 21.5 | <0.001 |
| | 25–34 | 704 | 48.8 | 127 332 | 68.4 | |
| | ≥35 | 84 | 5.8 | 18 928 | 10.2 | |
| Maternal height, cm | <150 | 9 | 0.6 | 465 | 0.3 | <0.001 |
| | 150–159 | 380 | 27.3 | 21 569 | 12.1 | |
| | 160–169 | 814 | 58.6 | 95 934 | 53.7 | |
| | ≥170 | 187 | 13.5 | 60 626 | 33.9 | |
| Missing | | | 3.7 | | 4.1 | |
| Maternal BMI, kg/m2 | < 18.5 | 136 | 10.0 | 5488 | 3.1 | <0.001 |
| | 18.5–24.9 | 748 | 54.8 | 110 845 | 63.2 | |
| | 25.0–29.9 | 329 | 24.1 | 40 412 | 23.0 | |
| | >30 | 153 | 11.2 | 18 743 | 10.7 | |
| Missing | | | 5.4 | | 5.8 | |
| Birthweight, g | ≤3500 | 937 | 65.0 | 94 218 | 50.6 | <0.001 |
| | 3501–4000 | 401 | 27.8 | 66 771 | 35.9 | |
| | ≥4001 | 103 | 7.1 | 25 183 | 13.5 | |
| Missing | | | 0.1 | | 0.1 | |
| Instrumental delivery | | 232 | 16.1 | 23 423 | 12.6 | <0.001 |
| Episiotomy in instrumental delivery | | 125 | 8.7 | 7159 | 3.8 | <0.001 |
| Episiotomy in all deliveries total | | 543 | 37.6 | 18 919 | 10.2 | <0.001 |
| Episiotomy, mediolateral | | 493 | 34.2 | 18 737 | 10.1 | <0.001 |
| Episiotomy, midline | | 47 | 3.3 | 136 | 0.1 | <0.001 |
| Obstetric anal sphincter injury | | 88 | 6.1 | 6514 | 3.0 | <0.001 |

*Fisher's exact test.

study was that the results were based on an assumption of FGM/infibulation prevalence due to the country of origin of the women rather than the actual reported diagnosis of FGM, which was the case in our study. Our results are also in accordance with the findings from a report from the National board of Health and Welfare [35], which concluded that women born in Africa, giving birth in Sweden, had the highest proportion of OASI (10%) followed by women born in Asia (7.8%), and in Sweden (5%).

To evaluate whether the risk of OASI could be influenced by country of origin, we performed a sensitivity analysis to evaluate whether the risk of OASI was remained when

**Table 2. Most frequent countries of origin based on the FGM group.**

| Country of origin | FGM n(%) | Non FGM n(%) |
|---|---|---|
| Somalia | 875 (60.6) | 986 (0.5) |
| Eritrea | 291 (20.2) | 1 488 (0.8) |
| Ethiopia | 88 (6.1) | 526 (0.3) |
| Sudan | 63 (4.4) | 150 (0.1) |
| Kenya | 22 (1.5) | 132 (0.1) |
| Gambia | 18 (1.2) | 98 (0.1) |
| Sweden | 18 (1.2) | 141 424 (75.9) |
| Other | 69 (4.8) | 41 490 (22.2) |

**Table 3. Association between obstetric anal sphincter injury (OASI), in relation to maternal and infant characteristics and obstetric interventions.**

| | All (n = 187 738) | Obstetric anal sphincter injury | | |
|---|---|---|---|---|
| | | n = 5702 | cOR (95% CI) | aOR (95% CI) |
| FGM status | | | | |
| Non FGM | 186 294 | 3.0% | 1 | 1 |
| FGM | 1 444 | 6.1% | 2.09 (1.68–2.60) | 2.69 (2.14–3.37) |
| Maternal age, years | | | | |
| <25 | 40 690 | 2.1% | 0.63 (0.58–0.68) | 0.67 (0.62–0.723) |
| 25–34 | 128 036 | 3.3% | 1 | 1 |
| ≥35 | 19 012 | 3.6% | 1.11 (1.03–1.21) | 1.06 (0.97–1.16) |
| Maternal height, cm | | | | |
| <150 | 474 | 3.0% | 0.95 (0.56–1.62) | |
| 150–159 | 21 949 | 3.8% | 1.23 (1.13–1.32) | |
| 160–169 | 96 748 | 3.1% | 1 | |
| ≥170 | 60 813 | 2.7% | 0.88 (0.83–0.94) | |
| Missing | 7754 | | | |
| Maternal BMI, kg/m2 | | | | |
| <18,5 | 5624 | 3.1% | 1.00 (0.86–1.17) | 1.17 (1.00–1.37) |
| 18.5–24.9 | 111 593 | 3.1% | 1 | 1 |
| 25–29.9 | 40 741 | 3.1% | 0.99 (0.93–1.06) | 0.93 (0.87–0.99) |
| ≥30 | 18 896 | 2.9% | 0.93 (0.85–1.02) | 0.86 (0.780–0.94) |
| Missing | 10 884 | | | |
| Birthweight, g | | | | |
| <3500 | 95 155 | 2.1% | 1 | 1 |
| 3500–4000 | 67 172 | 3.4% | 1.66 (1.56–1.76) | 1.67 (1.56–1.78) |
| ≥4001 | 25 286 | 5.8% | 2.90 (2.70–3.11 | 2.86 (2.66–3.07) |
| Missing | 125 | | | |
| Instrumental delivery | | | | |
| No | 164 083 | 2.5% | 1 | 1 |
| Yes | 23 655 | 7.0% | 2.96 (2.79–3.11) | 2.86 (2.66–3.07) |
| Episiotomy | | | | |
| No | 168 276 | 3.0% | 1 | 1 |
| Yes | 19 462 | 3.5% | 1.18 (1.09–1.28) | 0.76 (0.70–0.83) |

cOR- Crude odds ratio; aOR = Adjusted odds ratio (Maternal age, BMI, birthweight, instrumental delivery, episiotomy).

comparing risks of FGM in a sub cohort of women born in countries with high prevalence of FGM. The fact that the odds were slightly attenuated in the sub cohort compared to the main cohort indicates that some of the association between FGM and OASI is due to other factors associated with country of origin in addition to the FGM in itself. However, the fact that the

**Table 4. Crude and adjusted odds ratios (OR) and 95% confidence intervals (CI) for obstetric anal sphincter injury (OASI) comparing women with female genital mutilation (FGM) and women without FGM (ref).**

| | OR | 95% CI |
|---|---|---|
| Crude | 2.09 | 1.68–2.60 |
| Adjusted for age and BMI | 2.36 | 1.89–2.95 |
| Adjusted for age, BMI, birthweight and instrumental delivery | 2.52 | 2.01–3.16 |
| Adjusted for age, BMI, birthweight, instrumental delivery and episiotomy | 2.69 | 2.14–3.37 |

**Table 5. Crude and adjusted odds ratios (OR) and 95% confidence intervals (CI) for obstetric anal sphincter injury (OASI) comparing women with female genital mutilation (FGM) and women without FGM (ref) in a sub cohort of women born in FGM countries\*.**

|  | OR | 95% CI |
|---|---|---|
| Crude | 1.54 | 1.20–1.96 |
| Adjusted for age and BMI | 1.67 | 1.30–2.14 |
| Adjusted for age, BMI, birthweight and instrumental delivery | 1.62 | 1.25–2.09 |
| Adjusted for age, BMI, birthweight, instrumental delivery and episiotomy | 1.71 | 1.32–2.22 |

\*FGM countries according to the United Nations Children's Fund.

OR was still increased for women with FGM in this cohort suggests that a history of FGM in itself increases the risk of OASI.

A potential explanation for the attenuation in OR in the sub cohort compared to the main cohort could be that women originating from countries with a high frequency of FGM are similar in many other aspects such as sociocultural background and potential language difficulties, factors that might affect the outcome.

An interesting finding from our study was the increased association between FGM and episiotomy. Episiotomy is a surgical incision in the vaginal opening performed to enlarge the birth outlet to facilitate delivery. It is recommended to be performed on selective cases when hastening vaginal birth is needed or when clinical circumstance puts the patient at high risk of OASI. Large observational studies have shown that medio-lateral or lateral episiotomy can reduce the risk of OASI, both in vaginal and instrumental vaginal deliveries, when done selectively by indication [36–39]. Midline episiotomy, on the other hand, has shown to increase the risk of OASI and is not recommended [40]. The significantly higher proportion of episiotomies in women with FGM was in accordance with a large meta-analysis [12]. In our study, the overall episiotomy rates (all types) were higher in the FGM group compared to the non FGM group (37.6% vs 10.2%) and in particular midline episiotomy (3.3% vs 0.1%). Since midline episiotomy is not standard practice in obstetric care, we suspect that the higher reported proportion of this particular intervention in the FGM group might in fact represent defibulation (reopening of the sealed labias in type III). The ICD-code for defibulation (TLF00) was only introduced in 2019 in Sweden, which may have contributed to the misuse of the episiotomy code in lack of a defibulation code. However, from our data it is not clear whether the episiotomy represents a defibulation or an actual episiotomy where the muscles of the perineum are cut to facilitate delivery in obstetric emergencies, such as fetal distress or to prevent OASI.

Defibulation is a surgical procedure carried out to re-open the sealed labias that covers the vaginal introitus for women that have undergone type III FGM. It is performed with an incision of the midline scar tissue that covers the vaginal introitus and urethral opening. This can be done before or during pregnancy, as well as during delivery. The recommendation regarding its timing is inconclusive due to lack of evidence [10,41]. According to the WHO guidelines, it is recommended to perform defibulation for preventing and treating obstetric complications and the timing of the procedure should be based on the preference of the woman, access to health-care facilities, place of delivery and the health-care provider's skill level [41]. In our study, we lack information both regarding type of FGM and if defibulation have been performed or not.

Another interesting finding from our study was the increased association between FGM and instrumental delivery. Previous studies have reported conflicting results [11,12]. There can be both fetal and maternal indications for instrumental deliveries, however, there is no

plausible explanation why women with FGM have an increased association with instrumental delivery.

In Sweden,16 out of 21 health care-regions have specific guidelines for health care providers concerning FGM, however with inconsistent quality and variation. A few regions have guidelines regarding defibulation during pregnancy and childbirth [42]. Insufficient guidelines may contribute to regional differences in the obstetric treatment of these women [43].

Swedish healthcare professionals express inadequate knowledge about FGM and lack of technical skills concerning defibulation, which can contribute to the poor obstetric outcomes. Further, they find the interaction with the women complex, due to language barriers, cultural differences and due to the sensitivity of the issue of FGM itself [43–46]. Similar findings are also reported among healthcare professionals working in other countries [47–49]. Furthermore, it has been expressed that the insufficient documentation concerning FGM in medical records, often lead to stress for both the healthcare provider and the woman [44,47,50]. Several of these factors may also have influenced our results.

Data from specialized clinics in Australia [16] and Switzerland [17] show that women with FGM have similar obstetric outcomes to women without FGM, except for an increased risk for emergency caesarean section [16,17], and third degree perineal tears [17]. Both specialized care and defibulation have been suggested as key practices to prevent adverse obstetric outcomes.

The mechanism by which FGM poses a risk for OASI is unclear. Scar tissue from the procedure, leading to decreased tensile strength, has been discussed to be a plausible pathway for the risk of adverse obstetric outcome among women with a history of FGM [10,17,41]. However, few studies have focused on the degree of scar tissue and type of FGM. Considering our clinical experience with women with a history of FGM, we find it difficult to agree with previous theories that the scar tissue plays a major role on the effect of OASI, since scarring from cutting of labia and the clitoral glans should not have any major effect on the anal sphincter muscle.

The inelasticity in the scar tissue in type III, may play a role in anal sphincter tear, however defibulation, if needed, is practiced at latest in the crowning stage. Perhaps the type of FGM and the timing of defibulation are key contributors to the risk of OASI. There is a possibility that women with the diagnosis of FGM mostly had type III (the most obvious type), and that women with the less severe type were perhaps not diagnosed at all. Thus, having mostly women with the need of defibulation in the case group. Further, the timing of defibulation is not known. Perhaps performing the defibulation just as the head is crowning deprives the attention on communication, the speed of the birth of the baby and manual perineal support. Moreover, lack of knowledge among staff about FGM and inexperience of performing defibulation could cause stress for both women in delivery and the midwife, which also may play a role for the risk of OASI among women with FGM.

Perhaps the FGM, is only one of several factors contributing to the increased risk. We believe that inadequate knowledge and experience of women with FGM among healthcare professionals, language barriers, structural discrimination and lack of national guidelines may contribute to the poor obstetric outcomes in high income countries among women with a history of FGM. Based on the results from our study we suggest that national guidelines on obstetric management also includes specific awareness regarding the risk for OASI that women with a history of FGM have. In addition, these women should always be given the choice to undergo a defibulation antenatal or intrapartal. As health care providers we all need to work to prevent and protect women and girls from this harmful procedure but as long as it still occurs it is crucial to be aware of the potential complications that may result from FGM.

## Strength and limitations

The strength of our study is the large study population extracted from the MBR, a high quality register with high coverage and an extensive variable list. Another crucial factor increasing the validity of our result is that the cases were based on actual ICD-codes rather than assumptions of prevalence of FGM based on country of origin. Furthermore, our study focuses on primiparous women, the group with the highest risk of OASI.

A limitation of this study is misclassification of cases, due to insufficient reporting of FGM in medical records. The possible underreporting of FGM cases in the control group would probably lead to an underestimation of the risk of OASI associated with FGM in our study.

Information about the type of FGM was not reported in our data. However, our purpose was to explore the risk for primiparous women with a history of FGM, irrespective of type. As a suggestion for further studies, it would be of interest to review the medical journals for these patients to gain a deeper understanding of the type of FGM and whether the episiotomies were in fact potential defibulation or not.

## Conclusion

We demonstrated that women with a history of FGM had a significant increased risk of OASI in comparison with women without FGM when giving birth in a high-income delivery setting. It is crucial for health care providers to be aware of the possible consequences after FGM in order to prevent adverse obstetric outcome that might occur among women with a history of FGM.

## Acknowledgments

We would like to thank Hans Järnbert-Pettersson at the department for clinical science and education for statistical advice. We would also like to thank Cecilia Berger and Linnea Ladfors for supportive and fruitful discussions.

## Author Contributions

**Conceptualization:** Bita Eshraghi, Jonas Hermansson, Lena Marions.

**Data curation:** Bita Eshraghi, Jonas Hermansson, Lena Marions.

**Formal analysis:** Bita Eshraghi, Jonas Hermansson.

**Funding acquisition:** Lena Marions.

**Investigation:** Bita Eshraghi, Jonas Hermansson, Lena Marions.

**Methodology:** Bita Eshraghi, Jonas Hermansson.

**Project administration:** Bita Eshraghi, Jonas Hermansson, Lena Marions.

**Resources:** Bita Eshraghi, Lena Marions.

**Software:** Bita Eshraghi.

**Supervision:** Jonas Hermansson, Lena Marions.

**Validation:** Bita Eshraghi, Jonas Hermansson, Vanja Berggren, Lena Marions.

**Visualization:** Bita Eshraghi, Jonas Hermansson, Vanja Berggren, Lena Marions.

**Writing – original draft:** Bita Eshraghi.

**Writing – review & editing:** Bita Eshraghi, Jonas Hermansson, Vanja Berggren, Lena Marions.

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
