## [Decision Letter · Decision Letter 0]

11 Sep 2022

PONE-D-22-18105Risk of obstetric anal sphincter tear among primiparous women with a history of female genital mutilation, giving birth in SwedenPLOS ONE

Dear Dr. Eshraghi,

Thank you for submitting your manuscript to PLOS ONE. After careful consideration, we feel that it has merit but does not fully meet PLOS ONE’s publication criteria as it currently stands. Therefore, we invite you to submit a revised version of the manuscript that addresses the points raised during the review process.

Please note that we have only been able to secure a single reviewer to assess your manuscript. We are issuing a decision on your manuscript at this point to prevent further delays in the evaluation of your manuscript. Please be aware that the editor who handles your revised manuscript might find it necessary to invite additional reviewers to assess this work once the revised manuscript is submitted. However, we will aim to proceed on the basis of this single review if possible.

We look forward to receiving your revised manuscript.

Kind regards,

Callam Davidson

Editorial Office

PLOS ONE

Journal Requirements:

3. Please ensure that you include a title page within your main document. You should list all authors and all affiliations as per our author instructions and clearly indicate the corresponding author.

4. Please provide additional details regarding participant consent. In the ethics statement in the Methods and online submission information, please ensure that you have specified what type you obtained (for instance, written or verbal, and if verbal, how it was documented and witnessed). If your study included minors, state whether you obtained consent from parents or guardians. If the need for consent was waived by the ethics committee, please include this information.

5. Please include a sentence that describes the main limitation(s) of the study's methodology in your Abstract.

6. Please ensure that the study is reported according to the STROBE guideline, and include the completed STROBE checklist as Supporting Information. Please add the following statement, or similar, to the Methods: "This study is reported as per the Strengthening the Reporting of Observational Studies in Epidemiology (STROBE) guideline (S1 Checklist)."

Did your study have a prospective protocol or analysis plan? Please state this (either way) early in the Methods section. If a prospective analysis plan (from your funding proposal, IRB or other ethics committee submission, study protocol, or other planning document written before analyzing the data) was used in designing the study, please include the relevant prospectively written document with your revised manuscript as a Supporting Information file to be published alongside your study, and cite it in the Methods section. A legend for this file should be included at the end of your manuscript. 

If no such document exists, please make sure that the Methods section transparently describes when analyses were planned, and when/why any data-driven changes to analyses took place.

Reviewers' comments:

Reviewer's Responses to Questions

**Comments to the Author**

1. Is the manuscript technically sound, and do the data support the conclusions?

Reviewer #1: Yes

2. Has the statistical analysis been performed appropriately and rigorously? 

Reviewer #1: No

3. Have the authors made all data underlying the findings in their manuscript fully available?

Reviewer #1: No

4. Is the manuscript presented in an intelligible fashion and written in standard English?

Reviewer #1: Yes

5. Review Comments to the Author

Reviewer #1: Review on PONE-D-22-18105 "Risk of obstetric anal sphincter tear among primiparous women with a history of female genital mutilation, giving birth in Sweden".

The paper is well written and evaluates the risk of OASI in women with a history of FGM. There has previously been described an association between FGM and OASI, and it is important to further evaluate this risk in order to decrease the risk of OASI for these women. There are, however, some main concerns regarding the study. Sicnce the study includes a large amount of women in the cohort, there are a broader range of statistical possibilities. First, it is important to evaluate many different possible risk factors of OASI and including them in the multivariable analyses. The authors only include a few, and they do not report why they present differnet models. I would suggest that more variables are added (for example maternal age, ethnicity, etc) and that the authors evaluate the risk of interaction between the different factors. Especially, the interactions between FGM, OASI and episiotomy is interesting.

Further, it would be informative if the authros present the standard indications for episiotomy. And when and how is FGM diagnosed and registered, and when and how is OASI diagnosed and registered?

Are there any codes in the used dataset, that evaluates surgery prior to the delivery to enable vaginal birth without extensive tearing or cutting?

Based on the findings, do the authors suggest a modification of guiding women with a history of FGM regarding mode of delivery?

Line 284-285, the authors state that scarring af labia and the clitoral glance should not increase the risk of OASI. However, they do not come up with an alternative biomechanical explanation for the found association? What about the elasticity in scarred tissue? What about preventive strategies, like the Finnish grip, warm cloths etc. Please elaborate further on this.

6. PLOS authors have the option to publish the peer review history of their article (what does this mean?). If published, this will include your full peer review and any attached files.

Reviewer #1: No

---

## [Author Response · Author response to Decision Letter 0]

15 Oct 2022

Response to the reviewer (PLOS ONE)

Risk of obstetric anal sphincter tear among primiparous women with a history of female genital mutilation, giving birth in Sweden

Dear Editor

We appreciate you and the reviewer for your time in reviewing our paper and providing valuable comments. The authors have carefully considered the comments and tried our best to address every one of them. We hope that the manuscript after careful revisions meets your high standards. The authors welcome further constructive comments if any. Below we provide the point-by-point responses. All modifications in the manuscript have been highlighted in yellow.

Sincerely 

Bita Eshraghi, MD

1.Since the study includes a large amount of women in the cohort, there are a broader range of statistical possibilities. First, it is important to evaluate many different possible risk factors of OASI and include them in the multivariable analyses. The authors only include a few, and they do not report why they present different models. I would suggest that more variables are added (for example maternal age, ethnicity, etc) and that the authors evaluate the risk of interaction between the different factors. Especially, the interactions between FGM, OASI and episiotomy is interesting.

Thank you for your comment on this issue.

The risk factors for OASI have been discussed over time. We chose to use the risk factors with strongest evidence (line 94-98), that is, nulliparous, birthweight, instrumental delivery, ethnicity and in addition episiotomy. We had undermined the major risk factors for OASI and had only extracted those variables from the register.

As suggested, maternal age is included in all models in table 4 and 5. In a sensitivity analysis (table 5), women born in countries with high prevalence of FGM were examined separately to disentangle the potential confounding effect of country of origin on risk of our outcome. We chose to present table 4 and 5 with 4 different models to show the reader how the different variables affect the outcome. 

We have added a section regarding interactions in page 8, line 166-167 and page 12, line 195-197

2. Indication of Episiotomy. When and how is FGM diagnosed and registered, when and how is OASIS diagnosed

Thank you for pointing out the need for clarification regarding indication for episiotomy, and how and when OASI and FGM are diagnosed. This has given us the opportunity to improve the manuscript through revision. We have now added 3 sections in which we have explained this. 

Page 14, line 235-238. 

Page 6, lines 125-126.

Page 7, lines 146-149.

3. Are the any codes in the used dataset, that evaluates prior to the delivery to enable vaginal birth without extensive tearing or cutting

As described in line 248-250 the code for defibulation (TLF00) was first introduced in 2019 and since our data covers 2014-2018, we could not use this code.

4. Based on the finding, do the authours suggest a modification of guidling women with a history of FGM regarding mode of delivery?

Thank you for your comment.

Based on the results from our study we suggest that national guidelines on obstetric management also includes specific awareness regarding the risk for OASI that women with a history of FGM have. In addition, these women should always be given the choice to undergo a defibulation antenatal or intrapartal. 

We have added this paragraph to the discussion section on page 17, line number 314-318.

5. Line 284-285, the authors state that scarring of the labia and the clitoral glans should not increase the risk of OASI. However, they do not come up with an alternative biomechanical explanation for the found association? What about the elasticity in the scarred tissue? What about preventive strategies, like the Finnish grip, warm cloth etc. Please elaborate further on this.

Thank you for your comment on this issue. We have now a section in which we have explained this. Other preventive strategies such as Finnish grip or warm cloth are considered routine practice in Sweden and were not specifically registered in this study. 

Page 16-17, line number 298-309

In addition, we change the word risk to association since this would be the more correct term.

Line number: 234-235

Line number: 265-266

We also added a limitation in the abstract

Line: 38-39

We also did some clarification in data report on line number: 116

---

## [Decision Letter · Decision Letter 1]

15 Nov 2022

PONE-D-22-18105R1Risk of obstetric anal sphincter tear among primiparous women with a history of female genital mutilation, giving birth in SwedenPLOS ONE

Dear Dr. Eshraghi,

Thank you for submitting your manuscript to PLOS ONE. After careful consideration, we feel that it has merit but does not fully meet PLOS ONE’s publication criteria as it currently stands. Therefore, we invite you to submit a revised version of the manuscript that addresses the points raised during the review process.

We look forward to receiving your revised manuscript.

Kind regards,

Giovanni Buzzaccarini, M.D.

Academic Editor

PLOS ONE

Journal Requirements:

Additional Editor Comments:

Please, consider the reviewers' comments.

Reviewers' comments:

Reviewer's Responses to Questions

**Comments to the Author**

1. If the authors have adequately addressed your comments raised in a previous round of review and you feel that this manuscript is now acceptable for publication, you may indicate that here to bypass the “Comments to the Author” section, enter your conflict of interest statement in the “Confidential to Editor” section, and submit your "Accept" recommendation.

Reviewer #2: All comments have been addressed

Reviewer #3: (No Response)

2. Is the manuscript technically sound, and do the data support the conclusions?

Reviewer #2: Partly

Reviewer #3: Yes

3. Has the statistical analysis been performed appropriately and rigorously? 

Reviewer #2: Yes

Reviewer #3: Yes

4. Have the authors made all data underlying the findings in their manuscript fully available?

Reviewer #2: Yes

Reviewer #3: No

5. Is the manuscript presented in an intelligible fashion and written in standard English?

Reviewer #2: Yes

Reviewer #3: Yes

6. Review Comments to the Author

Reviewer #2: Dear authors,

I read with great interest the manuscript, which falls within the aim of this Journal. In my honest

opinion, the topic is interesting enough to attract the readers’ attention. Its very probably that women with a history of FGM had a significant increased risk of OASI in comparison with women without FGM when giving birth in a high income delivery setting.

Nevertheless, authors should clarify some points and improve the discussion, as suggested below. Authors should consider the following recommendations:

In my opinion auhtor's have to improve the paper refering in the text if the pregnancy came from IVF or natural conception and stress how is important the impact of assisted reproduction techniques on the neuro-psycho-motor outcome of newborns and aobut eventually congenital infections .

I suggest to read anf cite these articles:

Waterbirth: current knowledge and medico-legal issues

Congenital Zika Syndrome: Genetic Avenues for Diagnosis and Therapy, Possible Management and Long-Term Outcomes

Convalescent plasma use in pregnant patients with COVID-19 related ARDS: a case report and literature review

Neonatal Outcomes and Long-Term Follow-Up of Children Born from Frozen Embryo, a Narrative Review of Latest Research Findings

Impact of assisted reproduction techniques on the neuro-psycho-motor outcome of newborns: a critical appraisal

Reviewer #3: General Impression:

The authors conducted an interesting study to assess the risk of obstetric anal sphincter injury during normal labor in patients with a previous female genital mutilation. The research followed a robust methodology and statistical analysis, besides including a large sample size -which I believe to be representative. The manuscript is well-written, well-structured, include a comprehensive discussion of its strengths and limitations. I only have some minor comments to further improve the quality of the paper.

Comments:

1. I suppose the word “Centers” is missing from line 115 (from prenatal, delivery and neonatal care Centers”.

2. Regarding lines 133-136, I would suggest adding to the limitation that this coding weakness may lead to the inclusion of non-Swedish women who received vulvar surgery for precancerous indications, but have them classified as patients with FGM. This could lead to overestimation of the chance of having OASI in patients with FGM.

3. “Mode of delivery” is written twice in line 151. Please delete the repetition.

4. In line 159, please provide a reference that justifies the consideration of a p value equals to 0.05 as statistical significance difference,

5. There is an extra “and” in line 186. Please delete it.

6. Please delete lines 211-213 because it is a repetition of the previous two and a half lines.

7. Please include a data availability statement to comply with journal’s policy.

7. PLOS authors have the option to publish the peer review history of their article (what does this mean?). If published, this will include your full peer review and any attached files.

Reviewer #2: **Yes: **giuseppe gullo

Reviewer #3: **Yes: **Antoine Naem

---

## [Author Response · Author response to Decision Letter 1]

28 Nov 2022

Response to the reviewers (PLOS ONE)

Risk of obstetric anal sphincter tear among primiparous women with a history of female genital mutilation, giving birth in Sweden

Dear Editor

We appreciate you and the reviewers for your time in reviewing our paper and providing valuable comments. All authors have carefully considered the comments and have tried our best to address every one of them. We hope that the manuscript after our revisions meets your high standards, and we welcome further constructive comments if any. Below we provide the point-by-point responses. All modifications in the manuscript have been highlighted in yellow.

Sincerely 

Bita Eshraghi, MD

1.In my opinion authors have to improve the paper referring in the text if the pregnancy came from IVF or natural conception and stress how is important the impact of assisted reproduction techniques on the neuro-psycho-motor outcome of newborns and about eventually congenital infections. 

Thank you for your comment on this issue. Our main purpose has been to investigate the association between FGM and obstetric anal sphincter tear. New-borns neuro-psycho-motor outcome was not included in our analysis, and we also have no information about the conception mode or potential infections prior to delivery. Further, we do not have ethical permission to extract information regarding infections and mode of conception.

2. Regarding lines 133-136, I would suggest adding to the limitation that this coding weakness may lead to the inclusion of non-Swedish women who received vulvar surgery for precancerous indications but have them classified as patients with FGM. This could lead to overestimation of the chance of having OASI in patients with FGM.

The coding could cause an overestimate in the FGM group, however, the mean age of vulva cancer is around 70 years old, so the risk is very low among women in fertile age. Only 22 cases had the Z907 code in the cohort. 

There is however, un underestimation of FGM coding due to insufficient reporting of FGM in the medical records that probably leads to an underestimation of the risk of OASIS. 

4. In line 159, please provide a reference that justifies the consideration of a p value equals to 0.05 as statistical significance difference. 

The significance level var chosen because similar studies have used the same level of significance. Unfortunately, there was a mistake in the manuscript referring to P value being equal to 0.05. This in now corrected.

7. Please include a data availability statement to comply with journal’s policy

All data is available in the manucript. We added the p-value to line 196.

All other comments regarding missing words etc have been corrected.

---

## [Editor Report · Decision Letter 2]

5 Dec 2022

Risk of obstetric anal sphincter tear among primiparous women with a history of female genital mutilation, giving birth in Sweden

PONE-D-22-18105R2

Dear Dr. Eshraghi,

We’re pleased to inform you that your manuscript has been judged scientifically suitable for publication and will be formally accepted for publication once it meets all outstanding technical requirements.

Kind regards,

Giovanni Buzzaccarini, M.D.

Academic Editor

PLOS ONE

Additional Editor Comments (optional):

I thank you for having answered properly the reviewers comments.
---

## [Editor Report · Acceptance letter]

22 Dec 2022

PONE-D-22-18105R2 

Risk of obstetric anal sphincter tear among primiparous women with a history of female genital mutilation, giving birth in Sweden 

Dear Dr. Eshraghi:

I'm pleased to inform you that your manuscript has been deemed suitable for publication in PLOS ONE. Congratulations! Your manuscript is now with our production department. 

Kind regards, 

on behalf of

Dr. Giovanni Buzzaccarini 

Academic Editor

PLOS ONE